# Beyond Time-Average Convergence: Near-Optimal Uncoupled Online Learning via Clairvoyant Multiplicative Weights Update

**Georgios Piliouras**
SUTD
georgios@sutd.edu.sg

**Ryann Sim**
SUTD
ryann_sim@mymail.sutd.edu.sg

**Stratis Skoulakis**
EPFL
efstratios.skoulakis@epfl.ch

## Abstract

In this paper we provide a novel and simple algorithm, Clairvoyant Multiplicative Weights Updates (CMWU), for convergence to *Coarse Correlated Equilibria* (CCE) in general games. CMWU effectively corresponds to the standard MWU algorithm but where all agents, when updating their mixed strategies, use the payoff profiles based on tomorrow's behavior, i.e. the agents are clairvoyant. CMWU achieves constant regret of $\ln(m)/\eta$ in all normal-form games with m actions and fixed step-sizes $\eta$. Although CMWU encodes in its definition a fixed point computation, which in principle could result in dynamics that are neither computationally efficient nor uncoupled, we show that both of these issues can be largely circumvented. Specifically, as long as the step-size $\eta$ is upper bounded by $\frac{1}{(n-1)V}$, where $n$ is the number of agents and $[0, V]$ is the payoff range, then the CMWU updates can be computed linearly fast via a contraction map. This implementation results in an uncoupled online learning dynamic that admits a $O(\log T)$-sparse sub-sequence where each agent experiences at most $O(nV \log m)$ regret. This implies that the CMWU dynamics converge with rate $O(nV \log m \log T/T)$ to a CCE and improves on the current state-of-the-art convergence rate of *uncoupled online learning dynamics* [13, 1].

## 1 Introduction

The connection between online learning and game theory has been extensively studied for decades. The emergence of *online learning algorithms* guaranteeing comparable payoffs with the *highest-rewarding strategy* has provided a landmark method of understanding how selfish agents can act in an adversarial environment, while the notion of *Coarse Correlated Equilibrium* (CCE) has provided a game-theoretic characterization of the limiting behavior of such *online learning dynamics* [4, 20, 35, 21].

An important aspect of *online learning dynamics* is that agents can collectively learn a CCE without having explicit knowledge of the full game description. By definition, *online learning algorithms* take as input the sequence of vectors corresponding to the payoff of each possible action, making no assumption about how these payoff vectors are derived [22]. This idea of information exchange can be concisely described with the following distributed protocol, known in the literature as *uncoupled online learning dynamics* [11]:

1. No agent is aware of their payoff matrix or the payoff matrix of any other agent.

2. At each round $t$, each agent $i$ announces their mixed strategy $x_i^t$.

3. Based on the announced strategy vector $x^t := (x_1^t, \ldots, x_n^t)$, each agent $i$ **learns only** their payoff vector $u_i(x_{-i}^t)$.

36th Conference on Neural Information Processing Systems (NeurIPS 2022).

4. Each agent $i$ uses $(u_i(x^0_{-i}), \ldots, u_i(x^t_{-i}))$ to update their mixed strategy at round $t + 1$.

If agents use one of the classical *no-regret* online learning algorithms (e.g. Hedge, Regret Matching, Multiplicative Weights Update (MWU) etc), then the *time-average behavior* of the resulting uncoupled online learning dynamics will converge to CCE with rate $\Theta(1/\sqrt{T})$ [9]. Despite the fact that $\Theta(1/\sqrt{T})$ has proven to be the optimal time-average regret that an agent can achieve in the adversarial case, the question of determining which update rule gives the fastest convergence to CCE has remained open. Over the years, a series of works has proposed update rules with better and better rates [9, 12, 24, 33, 37, 10] until the recent seminal work of Daskalakis et. al [13], which established that *Optimistic Multiplicative Weights Update* (OMWU) admits $O(\log^4 T/T)$ rate of convergence. This rate matches (up to logarithmic factors) the lower bound $\Omega(1/T)$ on the convergence rate of any uncoupled online learning dynamic [11].

### Self-Play and Algorithmic Applications

Apart from theoretical interest, uncoupled online learning dynamics admit important applications which make the above line of research even more exciting and significant [35, 15, 25, 28, 39, 32, 27]. Some of their interesting algorithmic properties are: $i$) their decentralized nature permits efficient distributed implementation, $ii$) the implicit access to the game via payoff vectors permits game-abstractions with vastly reduced size, and $iii$) their iterative guarantees can be preferable over the *all-or-nothing guarantees* of linear programs. A notable example of their algorithmic success is the design of state-of-the art AI poker programs based on iterative self-play that outperform previous LP-based approaches, and were able to compete with human professionals [6, 38, 40]. We remark that this result has motivated a parallel line of research studying the convergence rates of uncoupled online learning dynamics for *extensive form games* [17, 8, 16, 18, 25].

### Our Contribution and Results

All previous works in this area couple the goals of minimizing adversarial regret and fast time-average convergence to CCE [9, 12, 24, 33, 15, 37, 10]. Since many of the aforementioned applications are in the realm of self-play where all agents are programmed to follow the update rule, guaranteeing adversarial no-regret is not a necessity. Indeed, it is not clear why time-average behavior should be the only way to deduce a CCE. In fact, any simple and efficient deduction rule would serve the algorithmic benefits of *uncoupled online learning dynamics* [18, 25]. Motivated by the above, our contributions can be summarized as follows:

- We introduce a novel update rule, called Clairvoyant Multiplicative Weights Update (CMWU) that produces sequences of strategy profiles with constant regret in general games.

- In its generic form, CMWU is a centralized update rule and does not fit in the online learning framework. However, based on CMWU we design an uncoupled online learning dynamic called CMWU dynamics which gives fast convergence to CCE *beyond the time-average sense*.

- More precisely, we establish that given any trajectory of length $T$ of CMWU dynamics, the $\log T$-sparse sub-trajectory always admits constant regret for all agents. As a result, the time-average behavior of the $O(\log T)$-sparse sub-trajectory converges to CCE with $\Theta(\log T/T)$ rate, improving on the previous state-of-the-art $\Theta(\log^4 T/T)$ achieved by Daskalakis et al. [13].

- The update rule of CMWU dynamics (presented in Algorithm 2) admits a simple form and an efficient implementation (requiring only a single step of the MWU algorithm at each round). The proof of convergence for CMWU dynamics is also far simpler than previous proofs of convergence of *uncoupled online learning dynamics* [13].

In Table 1 we summarize the most important results concerning the convergence to CCE of uncoupled online learning dynamics.

*Remark* 1.1. All the results mentioned in Table 1 additionally admit $\tilde{O}(\sqrt{T})$ adversarial guarantees. As mentioned above, this means that once a subset of the agents act adversarially by not following the update rule of the online learning dynamic, the agents following the update rule are guaranteed to experience at most $\tilde{O}(\sqrt{T})$ regret. The reason why CMWU is able to achieve better guarantees with simpler analysis comes from the fact that it neglects the adversarial no-regret guarantees that are

Table 1: Prior results for convergence to CCE in uncoupled online learning dynamics. $n$ denotes the number of players, $m$ denotes the number of actions per player and $V$ denotes the maximum value in the game payoff tensor. The step-size for CMWU, $\eta$, is set to $1/2Vn$.

| Update Rule | Deduction Rule | Rate of Convergence | Game Type |
|:---:|:---:|:---:|:---:|
| MWU | Time-average | $O\left(V\sqrt{\log m/T}\right)$ [9] | General-sum |
| Excessive Gap Technique | Time-average | $O\left(V\log m(\log T + \log^{3/2} m)/T\right)$ [12] | 2-player zero-sum |
| DS-OptMD, OptDA | Time-average | $\log^{O(V)}(m)/T$ [24] | 2-player zero-sum |
| OMWU | Time-average | $O\left(V\log m\sqrt{n}/T^{3/4}\right)$ [33, 37] | General-sum |
| OMWU | Time-average | $O\left(V\log^{5/6} m/T^{5/6}\right)$ [10] | General-sum |
| OMWU | Time-average | $O\left(nV\log m\log^4 T/T\right)$ [13] | General-sum |
| CMWU | $\log T-$sparse average | $O\left(nV\log m\log T/T\right)$ (**Theorem** 4.1) | General-sum |

irrelevant in self-play/training settings and focuses on minimizing the regret in specific parts of the sequence.

*Remark* 1.2. In another related work, [1] established an online learning dynamic that converges with rate $\Theta(n\log(m)\log^4 T/T)$ to *Correlated Equilibria* (CE), a subset of *Coarse Correlated Equilibria*. Finally, following our work, [2] produced time-average convergence to CE via online learning at a rate $\Theta(nm^{5/2}\log T/T)$. In comparison, our dependency on the number of actions $m$ is exponentially smaller at $\Theta(\log(m))$. Furthermore, their update rule is based on self-concordant regularization and thus admits high per-iteration complexity, a point which is mentioned by the authors.

*Remark* 1.3. Similarly to other standard no-regret algorithms, CMWU is designed to compute CCE via an uncoupled online learning dynamic and thus cannot be viewed as a rational behavioral assumption for selfish agents who seek to minimize their individual cost. One concrete example where the assumption fails is the case of auctions with a no-regret buyer [5], where the auctioneer can provably manipulate the behavior of any buyer learning according to a wide class of no-regret algorithms to guarantee themselves optimal revenue. This comes at the expense of higher prices for the buyer, since the auctioneer can take advantage of the commitment of the buyer to a no-regret learning strategy. Hence, it is clear that the learning in games community should discuss and differentiate in greater detail the uncoupled online learning dynamics that come as *natural game-play* and uncoupled dynamics that come as a means of *decentralized, efficient computation*.

**The Philosophy and Design of CMWU**

We introduce a radically different philosophy in the design of online learning algorithms. We shift away from the prevailing paradigm by defining a novel algorithm that we call *Clairvoyant Multiplicative Weights Update* (CMWU). CMWU is MWU equipped with a mental/simulated/synthetic model (jointly shared across all agents) about the state of the system in its next period. Each agent records its mixed strategy, i.e., its belief about what it expects to play in the next period in this shared mental model, which is internally updated using MWU without any changes to the real-world behavior up until it equilibrates, thus marking its consistency with the next day's real-world outcome. It is then and only then that agents take action in the real-world, effectively doing so with "full knowledge" of the state of the system on the next day, i.e., they are clairvoyant. CMWU acts as MWU with one day look-ahead, achieving bounded regret. CMWU update rule is closely related with the *Proximal Point Method* (PPM) [29, 26, 34, 31] which is an implicit (and therefore unimplementable) method that admits arbitrarily fast convergence in the context of convex minimization. Similarly to PPM, in order to implement the update rule of CMWU one would require access to the explicit description of the game and to additionally solve a *fixed-point problem*.

Our main technical contribution consists of establishing that for sufficiently small step-sizes, the update rule of CMWU can be computed via the iterations of a contraction map. This not only provides a way to compute the CMWU update rule using a simple and efficient process, but also opens a pathway for *decentralized computation* in the context of *uncoupled online learning dynamics*. The basic idea behind CMWU dynamics (Algorithm 2) is to punctuate the history of play with *anchor points* which are equally spaced in distance $\Theta(\log T)$, such that any anchor point is the CMWU update of the previous anchor point. In this way, the regret in the anchor sequence is constant for any agent. Moreover, the intermediate points between two anchor points correspond to the iterations of the contraction map. Surprisingly enough, all of the above can be implemented with just an *if condition* and an MWU-esque exponentiation (see Algorithm 2).

## 2 Preliminaries & Model

### 2.1 Normal Form Games

We begin with basic definitions from game theory. A finite normal-form game $\Gamma \equiv \Gamma(\mathcal{N}, \mathcal{S}, u)$ consists of a set of players $\mathcal{N} = \{1, ..., n\}$ where player $i$ may select from a finite set of actions or pure strategies $\mathcal{S}_i$. Each player has a payoff function $u_i : \mathcal{S} \equiv \prod_i \mathcal{S}_i \to \mathbb{R}$ assigning reward $u_i(s)$ to player $i$. It is common to describe $u_i$ with a payoff tensor $A^{(i)}$ where $u_i(s) = A_s^{(i)}$. Let $m = \max_i |\mathcal{S}_i|$ denote the maximum number of pure strategies in $\Gamma$, and let $V = \max_{i,s} |A_s^{(i)}|$ denote the maximum payoff value of any strategy in $\Gamma$.

Players are also allowed to use mixed strategies $x_i = (x_{is_i})_{s_i \in \mathcal{S}_i} \in \Delta(\mathcal{S}_i) \equiv \mathcal{X}_i$. The set of mixed strategy profiles is $\mathcal{X} = \prod_i \mathcal{X}_i$. A strategy is fully mixed if $x_{is_i} > 0$ for all $s_i \in \mathcal{S}_i$ and $i \in \mathcal{N}$. Individuals compute the payoff of a mixed strategy linearly using expectation. Formally,

$$u_i(x) = \sum_{s \in \mathcal{S}} u_i(s) \prod_{i \in \mathcal{N}} x_{is_i}. \tag{1}$$

We also introduce additional notation to express player payouts for brevity in our analysis later. Let $v_{is_i}(x) = u_i(s_i; x_{-i})$[1] denote the reward $i$ receives if $i$ opts to play pure strategy $s_i$ when everyone else commits to their strategies described by $x$. This results in $u_i(x) = \langle v_i(x_{-i}), x_i \rangle$. Next we will introduce the notion of Coarse Correlated Equilibria (CCE), which is the key equilibrium notion we explore in our work.

**Definition 2.1.** A probability distribution $\mu$ over pure strategy profiles $s = (s_1, \ldots, s_n) \in \mathcal{S}$ is called an $\epsilon$-*approximate Coarse Correlated Equilibrium* if for each agent $i \in [n]$,

$$\mathrm{E}_{s \sim \mu}[u_i(s)] \geq \mathrm{E}_{s \sim \mu}[u_i(s_i', s_{-i})] - \epsilon \quad \text{for all actions } s_i \in \mathcal{S}_i$$

**Definition 2.2.** Given a strategy profile $x := (x_1, \ldots, x_n) \in \mathcal{X}$, $\mu_x$ denotes the product probability distribution over strategy profiles $s = (s_1, \ldots, s_n) \in \mathcal{S}$ induced by $x$, $\mu_x(s) := \Pi_{s_i \in s} x_{is_i}$.

### 2.2 Uncoupled Online Learning Dynamics in Games

We study games from a learning perspective where agents iteratively update their mixed strategies over time based on the performance of pure strategies in prior iterations via an uncoupled, online adaptive algorithm. We will start by describing one of the most classical online learning algorithms, Multiplicative Weights Update (MWU).

The update rule for MWU can be written as

$$x_{is_i}^{t+1} = \frac{x_{is_i}^t \exp\left(\eta_i \cdot v_{is_i}(x^t)\right)}{\sum_{\bar{s}_i \in \mathcal{S}_i} x_{i\bar{s}_i}^t \exp\left(\eta_i \cdot v_{i\bar{s}_i}(x^t)\right)} \tag{MWU}$$

The remarkable guarantee of MWU and many other online learning algorithms is that the actual payoff for agents is close to the *highest-rewarding action* of the game. This property is formally captured via the notion of *regret*.

---

[1]$(s_i; x_{-i})$ denotes the strategy $x$ after replacing $x_i$ with $s_i$.

**Definition 2.3** (Regret). Given a sequence of mixed strategies $x_0, \ldots, x_{T-1}$ the regret of agent $i$, $R_i(T)$, is defined as

$$R_i(T) := \max_{x_i \mathcal{X}_i} \sum_{t=0}^{T-1} \langle v_i(x_{-i}), x_i^t \rangle - \sum_{t=0}^{T-1} \langle v_i(x_{-i}^t), x_i^t \rangle$$

Once agent $i$ adopts MWU, then they are ensured to experience sub-linear regret, $R_i(T) \leq O\left(\sqrt{T}\right)$ no matter how the other agents update their strategies [9]. This implies that the time-averaged difference between the *payoff of the best fixed strategy* and *the actual produced reward* goes to zero with rate $O(1/\sqrt{T})$. Any online learning algorithm that is able to guarantee $R_i(T) = o(T)$ is called *no-regret*.

There exists a folklore connection between any no-regret online learning algorithms and Coarse Correlated Equilibrium.

**Theorem 2.4** (Folkore[2]). *Given a sequence of mixed strategies $(x_0, \ldots, x_{T-1})$, then the probability distribution $\hat{\mu} := \sum_{t=0}^{T-1} \mu_{x_t}/T$ is an $(R(T)/T)$-approximate Coarse Correlated Equilibrium where $R(T) := \max_{i \in [n]} R_i(T)$.*

Theorem 2.4 implies that if all agents update their strategies with a *no-regret online learning algorithm*, then the time-average strategy vector converges to CCE with rate $o(T)/T$. As a result, the time-average strategy vector converges to CCE as $T \to \infty$.

In Algorithm 1, we present a generic description of uncoupled online learning dynamics.

---

**Algorithm 1** Uncoupled Online Learning Dynamics

---

1: **for all** round $t = 0, \cdots, T-1$ **do**
2:      Each player $i \in [n]$, **broadcasts** its mixed strategy $x_i^t \in \mathcal{X}_i$
3:      Each agent $i \in [n]$, **learns only** its reward vector $u_i(x_{-i}^t)$.
4:      Each agent $i \in [n]$, **updates** $x_i^{t+1} \in \mathcal{X}_i$ based only on $u_i(x_{-i}^0), \ldots, u_i(x_{-i}^t)$
5: **end for**

---

If for example the update rule of MWU is implemented at Step 4 of Algorithm 1, then the distribution $\hat{\mu} := \sum_{t=0}^{T-1} \mu_{x_t}/T$ is an $(1/\sqrt{T})$-approximate CCE.

## 3 Clairvoyant MWU

In this section, we introduce a novel learning algorithm for games that we call Clairvoyant Multiplicative Weights Updates (CMWU). Critically, CMWU, unlike MWU, *forms self-confirming predictions/beliefs* about what all opponents will play in the next time instance. Namely, all agents will form the same belief about what agent $i$ will play in the next period $t + 1$ $\left(x_i^{t+1}\right)$. These beliefs/estimates are such that when agents simulate an extra period of play in their mind and update their current strategies using MWU, the resulting strategy for each agent $i$ is $x_i^{t+1}$. All agents accurately *predict* the behavior of all other agents *tomorrow*, in other words they are *clairvoyant*!

The update rule for (CMWU) is as follows:

$$x_{is_i}^{t+1} = \frac{x_{is_i}^t \exp\left(\eta_i \cdot v_{is_i}(x^{t+1})\right)}{\sum_{\bar{s}_i \in \mathcal{S}_i} x_{i\bar{s}_i}^t \exp\left(\eta_i \cdot v_{i\bar{s}_i}(x^{t+1})\right)} \tag{CMWU}$$

CMWU is an implicit method: the new strategy $x^{t+1}$ appears on both sides of the equation, and thus the method needs to solve an algebraic/fixed point equation for the unknown $x^{t+1}$.

**Theorem 3.1.** *The algebraic system of equations in* (CMWU) *defined by an arbitrary game $\Gamma$, an arbitrary tuple of learning rates $\eta_i$, and any state $x^t$, always admits a solution.*

---

[2]See e.g. `https://theory.stanford.edu/~tim/f13/l/l17.pdf`

Theorem 3.1 follows directly by the Brouwer fixed-point theorem. Having established the fact that the agents can always *collectively compute* a next step of the CMWU update rule, we present the remarkable property of CMWU stated in Theorem 3.2.

**Theorem 3.2.** *Let $x_0, \ldots, x_{T-1}$ be a sequence of mixed strategies such that all pairs of consecutive mixed strategies ($x_t, x_{t+1}$) satisfy Equation* (CMWU). *Then, each agent $i$ has bounded regret $\leq \log|S_i|/\eta_i$.*

The proof of Theorem 3.2 follows by standard arguments in online learning literature (e.g. Lemma 5.4 in [22]). Moreover, we directly obtain the following corollary:

**Corollary 3.3.** *Let $x_0, \ldots, x_{T-1}$ be a sequence of mixed strategies such that all pairs of consecutive mixed strategies ($x_t, x_{t+1}$) satisfy Equation* (CMWU). *Then, the probability distribution $\hat{\mu} := \sum_{t=0}^{T-1} \mu_{x^t}/T$ is a $(\log m/\eta T)$-approximate CCE where $\eta := \min_{i \in n} \eta_i$.*

From a computational complexity perspective, solving the algebraic/fixed point equation (CMWU) is in general a hard problem. For instance, the computation of a Nash Equilibrium [30] which has proven to be PPAD-complete [14], reduces to the computation of a solution for equation (CMWU) once $\eta \to \infty$. Moreover, even if we assume that the agents possess unlimited computational power, it is not clear how they can compute a solution to Equation (CMWU) in the context of *uncoupled online learning dynamics*.

In Section 3.1, we show that if each $\eta_i$ is upper bounded by some game-dependent parameters, (CMWU) is a contraction map (Theorem 3.6, Corollary 3.7). This not only implies that there exists a unique fixed-point solution that can be computed very efficiently, but also that the Clairvoyant MWU update rule can be simulated via an uncoupled online learning dynamic.

## 3.1 Uniqueness of Fixed Point via Map Contraction

We will establish uniqueness of the fixed point in CMWU for a specific range of step-sizes. The proof will be based on an application of the Banach fixed-point theorem (mapping theorem or contraction mapping theorem) [3]. Thus, we simultaneously provide a constructive method to compute these fixed points with linear convergence rate. In what follows, it will be useful to consider (MWU) as a map from a vector of payoffs $v_i = (v_{i1}, \ldots, v_{i|S_i|})$ to mixed strategies parameterized by the current initial position $x_i^t$:

$$f_{x_i^t}(v_i) := \left( \frac{x_{i1}^t \exp\left(\eta_i \cdot v_{i1}\right)}{\sum_{\bar{s}_i \in \mathcal{S}_i} x_{i\bar{s}_i}^t \exp\left(\eta_i \cdot v_{i\bar{s}_i}\right)}, \ldots, \frac{x_{i|\mathcal{S}_i|}^t \exp\left(\eta_i \cdot v_{i|\mathcal{S}_i|}\right)}{\sum_{\bar{s}_i \in \mathcal{S}_i} x_{i\bar{s}_i}^t \exp\left(\eta_i \cdot v_{i\bar{s}_i}\right)} \right) \tag{MWU$_f$}$$

We first establish the fact that $f_{x_i^t}(v_i)$ is an $2\eta_i$-continuous mapping.

**Lemma 3.4.** *For any choice of $x_i^t \in \Delta(\mathcal{S}_i)$, the* (MWU$_f$) *map $f_{x_i^t} : \mathbb{R}^{|S_i|} \to \Delta(\mathcal{S}_i)$ satisfies that for any utility vectors $v_i, v_i' \in \mathbb{R}^{|S_i|}$,*

$$\|f_{x_i^t}(v_i) - f_{x_i^t}(v_i')\|_1 \leq 2\eta_i \|v_i - v_i'\|_\infty$$

In the rest of the section we establish that once all $\eta_i$ are selected sufficiently small then Equation MWU$_f$ admits a unique fixed point and is in fact a contraction map.

**Definition 3.5.** The distance between the strategy profiles $x = (x_1, \ldots, x_n) \in \mathcal{X}$ and $x' = (x_1', \ldots, x_n') \in \mathcal{X}$ is defined as $\mathcal{D}(x, x') := \max_{1 \leq i \leq n} \|x_i - x_i'\|_1$.

In Theorem 3.6 we show that the computation of CMWU is a contraction map once all $\eta_i$ do not exceed a game-dependent constant.

**Theorem 3.6.** *Consider the mixed strategy profile $(x_1^t, x_2^t, \ldots, x_n^t)$ and the map $G : \mathcal{X} \mapsto \mathcal{X}$ defined as follows:*

$$G(x) := \left( f_{x_1^t}(v_1(x)), \ldots, f_{x_n^t}(v_n(x)) \right)$$

*Then for any $x, x' \in \mathcal{X}$,*

$$\mathcal{D}(G(x), G(x')) \leq \eta V(n-1) \cdot \mathcal{D}(x, x')$$

*where $\eta$ is the maximum step-size over all players, $[0, V]$ is the payoff range of the game and $n$ is the number of players.*

*Proof.* Let us denote by $d_{\text{TV}}(x, x')$ the total variation distance between product distributions $x, x'$.

$$
\begin{aligned}
\mathcal{D}(G(x), G(x')) &= \|f_{x_i^t}(v_i(x)) - f_{x_i^t}(v_i(x'))\|_1 \quad \text{for some player } i \in [n] \\
&\leq 2\eta \cdot \|v_i(x) - v_i(x')\|_\infty \quad \text{by Lemma (3.4)} \\
&\leq 2\eta V \cdot d_{\text{TV}}(x_{-i}, x'_{-i}) \\
&\leq 2\eta V \cdot \sum_{j \neq i} d_{\text{TV}}(x_j, x'_j) \quad \text{by known properties of total variation (e.g., [23])} \\
&= \eta V \cdot \sum_{j \neq i} \|x_j - x'_j\|_1 \\
&\leq \eta V(n-1) \cdot \mathcal{D}(x, x')
\end{aligned}
$$

$\square$

**Corollary 3.7.** *The* $(\text{MWU}_f)$ *map with maximum step-size* $\eta < \frac{1}{(n-1)V}$ *is a contraction and thus converges to its unique fixed point at a linear rate.*

## 4 Uncoupled Clairvoyant MWU Online Learning Dynamics

In this section we present an uncoupled online learning dynamic based on the Clairvoyant MWU update rule, which we call CMWU dynamics. More precisely, the agents follow the distributed protocol described in Algorithm 1 while each agent $i$ runs Algorithm 2 internally to update their strategy $x_i^t$ at each round. To simplify notation in Algorithm 2, we set the number of actions of any agent $i$ to be $m := |\mathcal{S}_i|$.

---

**Algorithm 2** Internal update rule of Clairvoyant MWU Dynamics

1: **Input:** $\eta > 0$, $k \in \mathcal{N}$
2: $x_i^{-1} \leftarrow (1/m, \ldots, 1/m)$ and $z_i^{-1} \leftarrow (1/m, \ldots, 1/m)$
3: **for** each round $t = 0, \cdots, T-1$ **do**
4:      **if** $t \bmod k == 0$ **then**
5:          $x_i^t \leftarrow x_i^{t-1}$
6:          Agent $i$ **broadcasts** the mixed strategy $x_i^t$ and then **receives** the payoff vector $v_i(x_{-i}^t)$.
7:          Updates $z_i^t$ such that for all $s_i \in \mathcal{S}_i$,

$$
z_{is_i}^t \leftarrow \frac{z_{is_i}^{t-1} e^{\eta \cdot v_{is_i}(x_{-i}^t)}}{\sum_{\bar{s}_i \in \mathcal{S}_i} z_{i\bar{s}_i}^{t-1} e^{\eta \cdot v_{i\bar{s}_i}(x_{-i}^t)}}
$$

8:      **else**
9:          $z_i^t \leftarrow z_i^{t-1}$
10:         Updates $x_i^t$ such that for all $s_i \in \mathcal{S}_i$,

$$
x_{is_i}^t \leftarrow \frac{z_{is_i}^t e^{\eta \cdot v_{is_i}(x_{-i}^{t-1})}}{\sum_{\bar{s}_i \in \mathcal{S}_i} z_{i\bar{s}_i}^t e^{\eta \cdot v_{i\bar{s}_i}(x_{-i}^{t-1})}}
$$

11:         Agent $i$ **broadcasts** the mixed strategy $x_i^t$ and then **receives** the payoff vector $v_i(x_{-i}^t)$.
12:      **end if**
13: **end for**

---

**Theorem 4.1.** *Let* $x_0, \ldots, x_{T-1}$ *be the strategy vector once each agent internally adopts Algorithm 2 with* $\eta = 1/2nV$ *and* $k = \lceil \log T \rceil$*. Then for each agent* $i$*,*

$$
\sum_{\tau=0}^{T'} \left\langle v_i(x_{-i}^{k \cdot \tau}), x_i^{k \cdot \tau} \right\rangle - \max_{x_i \in \mathcal{X}_i} \sum_{\tau=0}^{T'} \left\langle v_i(x_{-i}^{k \cdot \tau}), x_i \right\rangle \geq -12nV \log m
$$

*where* $T' = \lfloor \frac{T-1}{k} \rfloor$*. Thus, the distribution* $\hat{\mu} := \sum_{\tau=0}^{T'} \mu_{x^{k\tau}}/T'$ *is a* $\Theta(nV \log m \log T/T)$*-approximate CCE.*

*Proof.* Notice that $z_i^t = z_i^{\lfloor t/k \rfloor}$ when $(t \bmod k) \neq 0$ and that

$$z_{is_i}^t \leftarrow \frac{z_{is_i}^{t-k} \cdot e^{\eta v_{is_i}(x_{-i}^t)}}{\sum_{\bar{s}_i \in \mathcal{S}_i} z_{i\bar{s}_i}^{t-k} \cdot e^{\eta v_{i\bar{s}_i}(x_{-i}^t)}} \quad \text{when } (t \bmod k) = 0 \tag{2}$$

As a result, the sequence $z_i^0, z_i^k, \dots, z_i^{k\tau}, \dots$ is the sequence produced by MWU with *look-ahead* (Be-The-Regularized-Leader) applied on the sequence reward vectors $v_i(x_{-i}^0), v_i(x_{-i}^k), \dots, v_i(x_{-i}^{k\tau}), \dots$ and it is known to admit the following regret guarantee[3],

$$\sum_{\tau=0}^{T'} \left\langle v_i(x_{-i}^{k\cdot\tau}), z_i^{k\cdot\tau} \right\rangle - \max_{x_i \in \mathcal{X}_i} \sum_{\tau=0}^{T'} \left\langle v_i(x_{-i}^{k\cdot\tau}), x_i \right\rangle \geq -\frac{\log m}{\eta} = -2nV \log m$$

Up next, we establish that $\|z_i^{k\cdot\tau} - x_i^{k\cdot\tau}\|_1 \leq 8/2^k$ for all $\tau \geq 1$. Let $\tau_m := (\tau - 1) \cdot k + m$. By the definition of Algorithm 2, $x_i^{k\cdot\tau} = x_i^{\tau_k - 1}$ and $z_i^{\tau_m} = z_i^{\tau_0}$ for all $m = 0, \dots, k-1$. Thus,

$$x_{is_i}^{\tau_m} \leftarrow \frac{z_{is_i}^{\tau_0} e^{\eta v_{is_i}(x_{-i}^{\tau_m - 1})}}{\sum_{\bar{s}_i \in \mathcal{S}_i} z_{i\bar{s}_i}^{\tau_0} e^{\eta v_{i\bar{s}_i}(x_{-i}^{\tau_m - 1})}} \quad \text{for all } s_i \in \mathcal{S}_i.$$

Using Equation $\text{MWU}_f$ of Section 3.1, the above system of equations can be concisely written as,

$$x_i^{\tau_m} = f_{z_i^{\tau_0}}\left( u_i(x_{-i}^{\tau_m - 1}) \right)$$

and since all agents follow Algorithm 2, $x^{\tau_m} = G\left( x^{\tau_m - 1} \right)$ where $G(x) := (f_{z_1^{\tau_0}}(x), \dots, f_{z_n^{\tau_0}}(x))$. Since $\eta := 1/2nV$ by Theorem 3.6, $G(x)$ is a contraction map with constant $1/2$. Thus,

$$\mathcal{D}\left( G\left( x^{\tau_k - 1} \right), x^{\tau_k - 1} \right) \leq \frac{1}{2^{k-2}} \cdot \mathcal{D}\left( x^{\tau_1}, x^{\tau_0} \right) \leq \frac{8}{2^k}.$$

The second inequality follows by $\mathcal{D}\left( x^{\tau_1}, x^{\tau_0} \right) \leq 2$ (see Definition 3.5). Recall that $x^{k\cdot\tau} = x^{\tau_k - 1}$ and that $z^{k\cdot\tau} = G\left( x^{\tau_k - 1} \right)$ (Equation 3). As a result, for each agent $i$

$$\|x_i^{k\cdot\tau} - z_i^{k\cdot\tau}\|_1 \leq \mathcal{D}\left( x^{k\cdot\tau}, z^{k\cdot\tau} \right) = \mathcal{D}\left( x^{\tau_k - 1}, G\left( x^{\tau_k - 1} \right) \right) \leq \frac{8}{2^k}$$

We are now ready to complete the proof of Theorem 4.1,

$$
\begin{aligned}
\sum_{\tau=0}^{T'} \langle v_i(x_{-i}^{k\cdot\tau}), x_i^{k\cdot\tau} \rangle \;&\geq\; \sum_{\tau=0}^{T'} \langle v_i(x_{-i}^{k\cdot\tau}), z_i^{k\cdot\tau} \rangle - |\langle v_i(x_{-i}^{k\cdot\tau}), x_i^{k\cdot\tau} - z_i^{k\cdot\tau} \rangle| \\
&\geq\; \sum_{\tau=0}^{T'} \langle v_i(x_{-i}^{k\cdot\tau}), z_i^{k\cdot\tau} \rangle - \sum_{\tau=0}^{T'} \|v_i(x_{-i}^{k\cdot\tau})\|_\infty \cdot \|x_i^{k\cdot\tau} - z_i^{k\cdot\tau}\|_1 \\
&\geq\; \sum_{\tau=0}^{T'} \langle v_i(x_{-i}^{k\cdot\tau}), z_i^{k\cdot\tau} \rangle - 8V\frac{T}{k2^k} - \|v_i(x_{-i}^0)\|_\infty \cdot \|x_i^0 - z_i^0\|_1 \\
&\geq\; \max_{x_i \in \mathcal{X}_i} \sum_{\tau=0}^{T'} \langle v_i(x_{-i}^{k\cdot\tau}), x_i \rangle - 2nV \log m - 10V \\
&\geq\; \max_{x_i \in \mathcal{X}_i} \sum_{\tau=0}^{T'} \langle v_i(x_{-i}^{k\cdot\tau}), x_i \rangle - 12nV \log m
\end{aligned}
$$

$\square$

---

[3]Lemma 5.4 in [22] or `https://web.stanford.edu/class/cs229t/scribe_notes/11_12_final.pdf`. For completeness we include the full proof in Lemma A.1.

# 5 Conclusion

In this paper we analyze a novel algorithm achieving the strongest to-date results for time-average convergence to coarse correlated equilibria (CCE) in general games. We call this Clairvoyant Multiplicative Weights Updates (CMWU). Although CMWU includes in its definition a fixed-point computation and effectively enables the agents to access the future behavior of their opponents, we show that it can be implemented efficiently in an online and totally uncoupled manner. A critical conceptual innovation of the uncoupled implementation of CMWU is that the CCE computation is done not via the standard techniques (uniform time averaging or last-iterate implementation) but an averaging of a pre-specified and game independent $O(\log T)$-sparse sub-sequence of the whole history of play. This type of implementation, as far as we know, is novel not only from a theoretical but even from a practical perspective.

Indeed, learning in games has become a fundamental as well as ubiquitous tool for numerous machine learning applications. These include the most well known AI headline success stories such as Generative Adversarial Networks [19], achieving superhuman performance in diverse settings such as Go [36], heads-up Poker [28], many-player Poker [7] a.o. Despite this wide range of settings, there is a roughly common pattern in achieving these results - design an optimization-driven learning dynamic and have it compete against itself in self-play. In poker style applications, the typical technique is to approximately solve the zero-sum game via low-regret *time-averaging* (typically uniform, sometimes with recency bias) over the *whole* history of play. In other applications such as Go, where the policy/strategy encoding is achieved by DeepNets with many parameters and any notion of averaging is totally impractical, the hope is that self-play leads to ever improving agents with the solution being the last-iterate of the self-play process. Our analysis shows that exploring different averaging techniques enables efficient, simple-to-implement, uncoupled and state-of-the-art algorithms to solving general normal-form games. This points to a rather under-explored hyper-parameter of online algorithm design, the *averaging policy*, and raises tantalizing questions about extending our results both in theory as well as experimentally to more complex settings.

## Acknowledgments

This research/project is supported by the National Research Foundation, Singapore and DSO National Laboratories under the AI Singapore Programme (AISG Award No: AISG2-RP-2020-016), NRF 2018 Fellowship NRF-NRFF2018-07, NRF2019-NRF-ANR095 ALIAS grant, grant PIE-SGP-AI-2020-01, AME Programmatic Fund (Grant No. A20H6b0151) from the Agency for Science, Technology and Research (A*STAR), Provost's Chair Professorship grant RGEPPV2101 and by EPSRC grant EP/R018472/1. Ryann Sim gratefully acknowledges support from the SUTD President's Graduate Fellowship (SUTD-PGF).

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
