# Appendix

## A Omitted Proofs

**Lemma A.1.** *Consider $z^{-k} = (1/m, \ldots, 1/m)$, the sequence of mixed strategies $z_i^0, z_i^k, \ldots, z_i^{kT'}$ and the sequence of reward vectors $v_i(x_{-i}^0), v_i(x_{-i}^k), \ldots, v_i(x_{-i}^{kT'})$ such that*

$$z_{is_i}^{k\cdot\tau} \leftarrow \frac{z_{is_i}^{k\cdot\tau-k} \cdot e^{\eta v_{is_i}(x_{-i}^{k\cdot\tau})}}{\sum_{\bar{s}_i \in \mathcal{S}_i} z_{i\bar{s}_i}^{k\cdot\tau-k} \cdot e^{\eta v_{i\bar{s}_i}(x_{-i}^{k\cdot\tau})}} \quad \text{for all } \tau \geq 0$$

*Then, the following guarantee holds,*

$$\sum_{\tau=0}^{T'} \left\langle v_i(x_{-i}^{k\cdot\tau}), z_i^{k\cdot\tau} \right\rangle - \max_{x_i \in \mathcal{X}_i} \sum_{\tau=0}^{T'} \left\langle v_i(x_{-i}^{k\cdot\tau}), x_i \right\rangle \geq -\frac{\log m}{\eta}.$$

*Proof.* To simplify notation let $t := k \cdot \tau$ and $v_i^t := v_i(x_{-i}^{k\cdot\tau})$. It is known that the $z_i^t$ can be equivalently described as (see Section 5.4.1 in [22]),

$$z_i^t = \arg\max_{z_i \in \mathcal{X}_i} \left[ \gamma \left\langle \sum_{s=0}^{t} u_i^s, z_i \right\rangle - h(z_i) \right] \text{ for all } t \geq -1$$

where $h(z_i) = -\sum_{s_i} z_{is_i} \log z_{is_i}$. Now let $g_t(z_i) := \gamma \left\langle \sum_{s=0}^{t} u_i^s, z_i \right\rangle - h(z_i)$ which means that $z_i^t = \arg\max_{z_i \in \mathcal{X}_i} g_t(z_i)$ and let $x_i^* := \arg\max_{x_i \in \mathcal{X}_i} \sum_{t=0}^{T'} \langle v_i^t, x_i \rangle$. Using a simple induction (see Lemma 5.4 in [22]) one can easily show that

$$\sum_{t=-1}^{T'} g_t(z_i^t) \geq \sum_{t=-1}^{T'} g_t(x_i^*)$$

which implies that

$$\sum_{\tau=0}^{T'} \left\langle v_i(x_{-i}^t), z_i^t \right\rangle - \sum_{\tau=0}^{T'} \left\langle v_i(x_{-i}^t), x_i^* \right\rangle \geq \frac{h(z^{-1})}{\gamma} - \frac{h(x_i^*)}{\gamma} \geq \frac{\log m}{\gamma}$$

$\square$

## B Proof of Lemma 3.4

To simplify notation we drop the dependence on $x_i^t$ and denote with $f_{s_i}(v_i)$ the $s_i$ coordinate of $f_{x_i^t}(v_i)$. Notice that for any $s_i, s_i' \in S_i$ with $s_i \neq s_i'$ we have:

$$\frac{\partial f_{s_i}}{\partial v_{is_i}} = \eta_i \frac{x_{is_i}^t \exp\left(\eta_i \cdot v_{is_i}\right)\left(\sum_{\bar{s}_i \in \mathcal{S}_i} x_{i\bar{s}_i}^t \exp\left(\eta_i \cdot v_{i\bar{s}_i}\right)\right)}{\left(\sum_{\bar{s}_i \in \mathcal{S}_i} x_{i\bar{s}_i}^t \exp\left(\eta_i \cdot v_{i\bar{s}_i}\right)\right)^2} - \frac{\left(x_{is_i}^t \exp\left(\eta_i \cdot v_{is_i}\right)\right)^2}{\left(\sum_{\bar{s}_i \in \mathcal{S}_i} x_{i\bar{s}_i}^t \exp\left(\eta_i \cdot v_{i\bar{s}_i}\right)\right)^2}$$

$$= \eta_i x_{is_i}^{t+1}(1 - x_{is_i}^{t+1})$$

Moreover,

$$\frac{\partial f_{s_i}}{\partial v_{is_i'}} = -\eta_i \frac{x_{is_i}^t \exp\left(\eta_i \cdot v_{is_i}\right) x_{is_i'}^t \exp\left(\eta_i \cdot v_{is_i'}\right)}{\left(\sum_{\bar{s}_i \in \mathcal{S}_i} x_{i\bar{s}_i}^t \exp\left(\eta_i \cdot v_{i\bar{s}_i}\right)\right)^2}$$

$$= -\eta_i x_{is_i}^{t+1} x_{is_i'}^{t+1}$$

It follows that $\|\nabla f_{s_i}(v_i)\|_1 = 2\eta_i x_{is_i}^{t+1} \cdot (1 - x_{is_i}^{t+1}) \le 2\eta_i x_{is_i}^{t+1}$ and thus $\sum_{s_i \in \mathcal{S}_i} \|\nabla f_{s_i}(v_i)\|_1 \le 2\eta_i$.

$$
\begin{aligned}
\|f_{x_i^t}(v_i) - f_{x_i^t}(v_i')\|_1 &= \sum_{s_i \in \mathcal{S}_i} |f_{s_i}(v_i) - f_{s_i}(v_i')| \\
&= \sum_{s_i \in \mathcal{S}_i} |\int_{t=0}^1 \langle \nabla f_{s_i}((1-t)v_i + tv_i'), v_i - v_i' \rangle \partial t| \\
&\le \sum_{s_i \in \mathcal{S}_i} \int_{t=0}^1 |\langle \nabla f_{s_i}((1-t)v_i + tv_i'), v_i - v_i' \rangle| \partial t \\
&\le \int_{t=0}^1 \Big( \sum_{s_i \in \mathcal{S}_i} \|\nabla f_{s_i}((1-t)v_i + tv_i')\|_1 \Big) \cdot \|v_i - v_i'\|_\infty \partial t \\
&\le 2\eta_i \|v_i - v_i'\|_\infty
\end{aligned}
$$

## C  Experimental Results

In this section we present several experimental results that show the fast convergence of CMWU to CCE. In Figure 1 we compare the performance of CMWU dynamics (Algorithm 2) to the current state of the art OMWU with step-sizes selected according to [13]. The game is a randomly generated 4-player, 10-strategy normal form game and in each run, the players' initial conditions are randomly generated. The update rule for (OMWU), also referred to as Optimistic Hedge, can be written as

$$
x_{is_i}^{t+1} = \frac{x_{is_i}^t \exp\left(\eta_i \cdot \left(2v_{is_i}(x^t) - v_{is_i}(x^{t-1})\right)\right)}{\sum_{\bar{s}_i \in \mathcal{S}_i} x_{i\bar{s}_i}^t \exp\left(\eta_i \cdot \left(2v_{i\bar{s}_i}(x^t) - v_{i\bar{s}_i}(x^{t-1})\right)\right)} \tag{OMWU}
$$

In order to account for the internal update rule of CMWU dynamics, we run the OMWU experiment for a longer time and compute the regret of the OMWU dynamic only at each $\log(T)$-th iterate. We observe that CMWU allows for faster computation of CCE than OMWU.

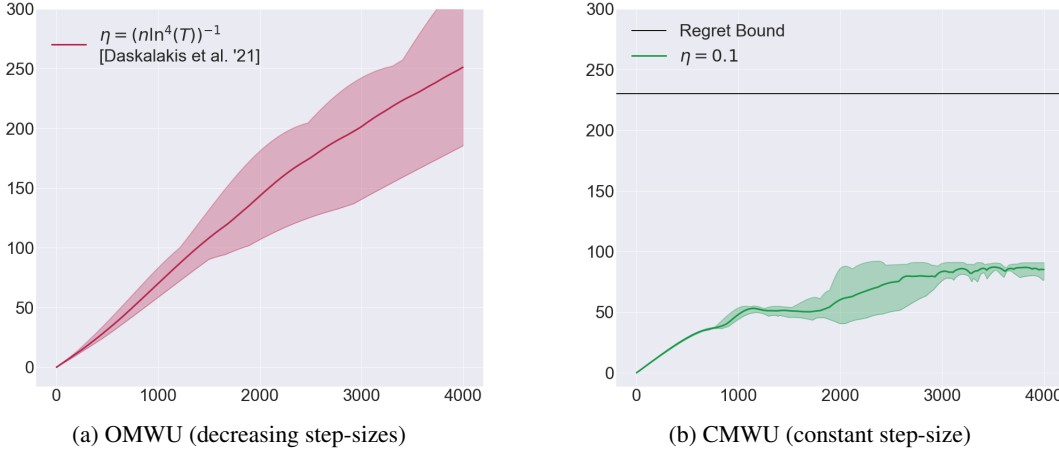

(a) OMWU (decreasing step-sizes)    (b) CMWU (constant step-size)

Figure 1: State-of-the-art OMWU [13] vs. CMWU in a 4-player 10-strategy game. We plot the max over agents' cumulative regret for several common random initializations. The shaded region represents the max/min regret range across runs. CMWU allows for significantly faster computation of approximate coarse correlated equilibria than OMWU, i.e., it needs significantly less oracles call for the same accuracy level. For a zoom-in on the cumulative regret of CMWU for larger step-sizes $\eta$ see Fig. 2.

In Figure 2 we plot the cumulative regret of CMWU for various fixed step-size values. As the step-size increases, we note empirically that the time required to compute a CCE decreases.

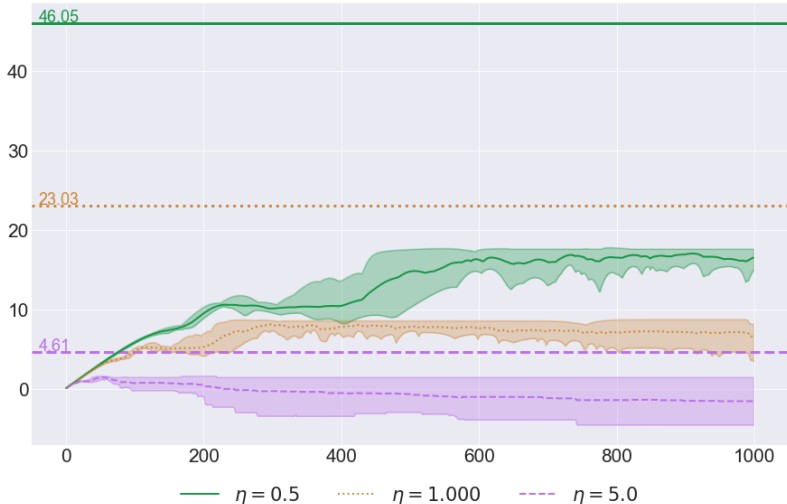

Figure 2: Zoomed-in cumulative regret over time for Figure 1b with larger $\eta$ values. As the learning rate increases, antithetical with other approaches, the speed of convergence of CMWU to CCE increases. Colored horizontal lines represent the respective theoretical regret bounds for each value of $\eta$.

## D    CMWU Dynamics as an Anytime Algorithm

Our formulation of the internal update rule of CMWU dynamics in Algorithm 2 can also be framed as an anytime algorithm. In this setting, the time horizon $T$ is not known in advance and thus the algorithm has to have bounded regret for all $T$. Typically one can obtain such an anytime algorithm via a doubling trick, but we propose a simple modification of the internal update rule which achieves the same effect in Algorithm 3. Our convergence result of Theorem 4.1 can also be extended to the anytime setting, as we show in Theorem D.1.

**Theorem D.1.** *Let $x_0, \ldots, x_{T-1}$ be the strategy vector once each agent internally adopts Algorithm 3 with $\eta = 1/2nV$. Then for each agent $i$,*

$$\sum_{\tau \in T'} \left\langle v_i(x^\tau_{-i}), x^\tau_i \right\rangle - \max_{x_i \in \mathcal{X}_i} \sum_{\tau \in T'} \left\langle v_i(x^\tau_{-i}), x_i \right\rangle \geq -O(nV \log m)$$

*Moreover $|T'| = \Omega(T/\log T)$ and thus the distribution $\hat{\mu} := \sum_{\tau \in T'} \mu_{x^\tau}/T'$ is a $O\left(nV \log m \log T/T\right)$-approximate CCE.*

*Proof.* Notice that the set $T'$ is the same for any agent $i$. In order to simplify notation let $T' = \{1, \ldots, \tau_{k-1}, \tau_k, \ldots, \tau_K\}$. At the same time, note that if Algorithm 3 is run for $T$ time-steps, then $K = \Omega(T/\log T)$. As in the proof of Theorem 3.2 we have that for any $\tau_k \in T'$, by definition of Algorithm 3,

$$\left| x^{\tau_k}_{is_i} - \frac{x^{\tau_{k-1}}_{is_i} \cdot e^{\eta v_{is_i}(x^{\tau_{k-1}}_{-i})}}{\sum_{\bar{s}_i \in \mathcal{S}_i} x^{\tau_{k-1}}_{is_i} \cdot e^{\eta v_{i\bar{s}_i}(x^{\tau_{k-1}}_{-i})}} \right| \leq \frac{1}{2^{\tau_k - \tau_{k-1}}} \leq \frac{1}{k^2} \tag{3}$$

To simplify notation we rewrite the above inequality as

$$\|x^{\tau_k} - y^{\tau_k}\| \leq 1/k^2$$

where

$$y^{\tau_k}_{is_i} \leftarrow \frac{x^{\tau_{k-1}}_{is_i} \cdot e^{\eta v_{is_i}(x^{\tau_{k-1}}_{-i})}}{\sum_{\bar{s}_i \in \mathcal{S}_i} x^{\tau_{k-1}}_{is_i} \cdot e^{\eta v_{i\bar{s}_i}(x^{\tau_{k-1}}_{-i})}}$$

---
**Algorithm 3** Anytime internal update rule of Clairvoyant MWU Dynamics
---
1: **Input:** $\eta > 0$
2: $x_i^0 \leftarrow (1/m, \ldots, 1/m)$ and $z_i^0 \leftarrow (1/m, \ldots, 1/m)$
3: $T' \leftarrow \{1\}$ and $\tau \leftarrow 0$
4: **for** each round $t = 1, \cdots, T - 1$ **do**
5:      **if** $t == \tau + \log\left(|T'|^2\right)$ **then**
6:          $x_i^t \leftarrow x_i^{t-1}$
7:          Agent $i$ **broadcasts** the mixed strategy $x_i^t$ and then **receives** the payoff vector $v_i(x_{-i}^t)$.
8:          Updates $z_i^t$ such that for all $s_i \in \mathcal{S}_i$,

$$z_{is_i}^t \leftarrow \frac{z_{is_i}^{t-1} e^{\eta \cdot v_{is_i}(x_{-i}^t)}}{\sum_{\bar{s}_i \in \mathcal{S}_i} z_{i\bar{s}_i}^{t-1} e^{\eta \cdot v_{i\bar{s}_i}(x_{-i}^t)}}$$

9:          $T' \leftarrow T' \cup \{t\}$ and $\tau \leftarrow t$
10:      **else**
11:          $z_i^t \leftarrow z_i^{t-1}$
12:          Updates $x_i^t$ such that for all $s_i \in \mathcal{S}_i$,

$$x_{is_i}^t \leftarrow \frac{z_{is_i}^t e^{\eta \cdot v_{is_i}(x_{-i}^{t-1})}}{\sum_{\bar{s}_i \in \mathcal{S}_i} z_{i\bar{s}_i}^t e^{\eta \cdot v_{i\bar{s}_i}(x_{-i}^{t-1})}}$$

13:          Agent $i$ **broadcasts** the mixed strategy $x_i^t$ and then **receives** the payoff vector $v_i(x_{-i}^t)$.
14:      **end if**
15: **end for**
---

The proof is completed with the exact same argument as in Theorem 3.2. More precisely,

$$
\begin{aligned}
\sum_{k=1}^{K} \langle v_i(x_{-i}^{\tau_k}), x_i^{\tau_k} \rangle &\geq \sum_{k=1}^{K} \langle v_i(x_{-i}^{\tau_k}), y_i^{\tau_k} \rangle - |\langle v_i(x_{-i}^{\tau_k}), x_i^{\tau_k} - y_i^{\tau_k} \rangle| \\
&\geq \sum_{k=1}^{K} \langle v_i(x_{-i}^{\tau_k}), y_i^{\tau_k} \rangle - \sum_{k=1}^{K} \|v_i(x_{-i}^{\tau_k})\|_\infty / k^2 \\
&\geq \sum_{k=1}^{K} \langle v_i(x_{-i}^{\tau_k}), y_i^{\tau_k} \rangle - O(V) \\
&\geq \max_{x_i \in \mathcal{X}_i} \sum_{k=1}^{K} \langle v_i(x_{-i}^{\tau_k}), x_i \rangle - O(nV \log m)
\end{aligned}
$$

$\square$