# OpenReview forum: "Beyond Time-Average Convergence: Near-Optimal Uncoupled Online Learning via Clairvoyant Multiplicative Weights Update"
_NeurIPS.cc/2022/Conference — NeurIPS 2022 Accept_

### Official Review · Reviewer_CoXh · 2022-07-03

**Rating:** 7
**Confidence:** 4
**Soundness:** 3 good
**Presentation:** 3 good
**Contribution:** 4 excellent

**Summary:**

The authors propose a Clairvoyant Multiplicative Weights Updates (CMWU) for learning normal-form games. The new method achieves the optimal regret with constant stepsize. The authors also provide an efficient decentralized implementation of the proposed method, which enjoys the optimal regret for learning a CCE.

**Questions:**

- Why are regret bounds in Table 1 called **Rate of Convergence**?

- References about Theorem 2.4 would be helpful.

- Do you have experimental justification of improvements shown in Table 1? It would be very helpful.

- Do you assume that the number of players is fixed? Is it restrictive?

- A typo in line 159: implementatio

**Limitations:**

Yes.

**Strengths And Weaknesses:**

## Originality

- The proposed CMWU is new and it improves existing regret bounds that have been compared.

- The proof has several novel ingredients, e.g., contraction property of MWU mapping.

- An uncoupled implementation of CMWU is a new decentralized multi-agent online learning algorithm.

## Quality & Clarity

- The paper is well written and all claims are justified via proofs.

## Significance

- The proposed CMWU takes a nice decentralized online implementation which is important for online multi-agent game learning.

- The defined regret bound measures performance via sparse part of history instead of time average of history. This goes beyond the standard metric in learning games.

- The established regret bound improves the dependence on $T$ compared with the existing methods. This is an important advance in learning normal form games.

---

> ### Author Response · Authors · 2022-08-01
> **Response to Reviewer CoXh**
>
> Thank you for your review and support of our paper. Please find the responses to your questions below.
> > Why are regret bounds in Table 1 called Rate of Convergence?}
>
> Since we focus on the convergence properties to CCE (which can be achieved with different deduction rules) we think it is more descriptive to show the rate of convergence to CCE. In the camera ready version, we can also add the regret bounds (for all the previous works this is simply Rate of Convergence $\times$ T).
>
> > References about Theorem 2.4 would be helpful.}
>
> We will add a reference.
>
>
> > Do you have experimental justification of improvements shown in Table 1? It would be very helpful.
>
> We have conducted experiments showing that CMWU appears to have superior experimental evaluations compared to OWMU dynamics. Since the scope of the paper is theoretical, we chose not to include them but we can add them to the camera ready version.
>
> > Do you assume that the number of players is fixed? Is it restrictive?
>
> This is a very interesting question. If agents are added or removed from the system in multiples of $k$, then CMWU computes a CCE for such time-varying game with $\log T /T $ rate. This is an interesting by-product contribution of our work and we will certainly discuss the above observation in detail in the camera-ready version. We remark that OMWU dynamics do not admit this property.

---

### Official Review · Reviewer_UCa9 · 2022-07-10

**Rating:** 6
**Confidence:** 4
**Soundness:** 3 good
**Presentation:** 3 good
**Contribution:** 3 good

**Summary:**

The paper introduces an uncoupled learning algorithm with efficient strategy updates, coined Clairvoyant Multiplicative Weights Update (CMWU). The algorithm produces a sequence of strategies, a subsequence of which minimizes regret.

The main idea behind CMWU stems from the known fact that by having a perfect (clairvoyant) prediction of the next loss vector in optimistic MWU, the regret of optimistic MWU grows constant. The contribution of the paper is in showing that such clairvoyant predictions can be obtained efficiently through fixed-point iterations of a contraction, and show that such fixed-point iterations can be interpreted as additional (but ignored in the computation of regret) iterates of a learning algorithm.

**Questions:**

Please feel free to push back on my analysis of weaknesses above, I'm certainly open to change my mind.

Beyond that, I have a few questions:
1. Other works in this space (e.g., the recent paper by Daskalakis et al. on near-optimal regret in games) stressed how the accelerated regret bounds can fall back to the usual, non accelerated, $O(\sqrt T)$ regret dynamics when facing adversarial opponents. Is the same possible for CMWU?

2. Can CMWU be run as an anytime algorithm, that is, without knowing the value of $T$ a priori? If so, I find that that would be an important addition to the paper.

**Limitations:**

Please see Strengths and Weaknesses above for a description of what limitations where not addressed adequately by the authors.

**Strengths And Weaknesses:**

While the paper relies on a simple and well-known observation---namely that the observation that clairvoyance leads to constant regret in MWU, which can be already found in these notes from 2008 (Section 2.3, [1])---and the proof of the contraction is incredibly elementary, I think the final result is noteworthy and deserves recognition. Indeed, I think that in this case the simplicity of the argument is a strength and not a weakness, and it is perhaps even surprising that this result had not been noted before.

I find the biggest limitation of the work to be the fact that the iterates produced by the proposed algorithm, CMWU, do not minimize regret. Rather, only a specific (log T)-spaced subsequence of the iterates do. It is not known whether the full sequence of iterates (and not only the specific subsequence) is regret minimizing. For that reason, I find calling CMWU an "algorithm for regret minimization" (abstract, line 2) misleading, especially given that the paper insists on the algorithm being uncoupled. I would encourage the authors to reword the abstract to avoid the confusion.

Continuing on the previous points, the authors call the fact that the dynamics only minimize regret when restricting to a specific (log T)-spaced subsequence of the iterates a "critical conceptual innovation" (line 255-256). On line 59-60, the authors try to back up the innovation by saying that "it is not clear why time-average behavior should be the only way to deduce a CCE". I only partially agree with this. I certainly agree that if the goal is CCE computation, then an alternative deduction rule that preserves uncoupledness and efficient strategy updates remains interesting (and in fact many alternative deduction rules have been tried in the literature, for instance alternative averaging of iterates are very popular in the extensive-form game community). Yet, assuming that dynamics are not run with the objective of computing a CCE, and rather only to guarantee some form of rational (i.e., regret-minimizing in this context) play/learning between the agents, the fact that the iterates as a whole do not minimize regret is clearly a downside. The paper seems to dance between these two intertwined but different goals---computing a CCE and defining regret-minimizing online play/learning---and so a clear discussion of the limitations of the alternative deduction rule (good for one scenario but not the other) gets missed. I strongly encourage the authors to elaborate on the downsides of the lack of the regret minimization property (at least in the usual sense) of their algorithm, and what problems remain open.


[1] http://www.mit.edu/~rakhlin/papers/online_learning.pdf

---

> ### Author Response · Authors · 2022-08-01
> **Response to Reviewer UCa9**
>
> Thank you for your review and feedback, please find our responses to your questions below.
>
> > I find the biggest limitation of the work to be the fact that the iterates produced by the proposed algorithm, CMWU, do not minimize regret...
>
> We will be happy to rephrase the abstract to avoid any possibility of confusion. However we remark that the update rule of CMWU (Algorithm $2$) is an online learning algorithm (despite the fact that it may not guarantee sublinear adversarial regret) and thus CMWU dynamics is a proper uncoupled online learning dynamic.
>
> > Continuing on the previous points, the authors call the fact that the dynamics only minimize regret when restricting to a specific (log T)-spaced subsequence of the iterates a "critical conceptual innovation" (line 255-256) ... I strongly encourage the authors to elaborate on the downsides of the lack of the regret minimization property (at least in the usual sense) of their algorithm, and what problems remain open.
>
> The scope of our work is the computation of CCE through an uncoupled online learning dynamic. The algorithmic merits of equilibrium computation through uncoupled online learning dynamics are exhibited and is the main motivation behind the long line of research studying online learning in games [Daskalakis et al, '21]. As a result, in this context any deduction rule that figures out a CCE ($\log T$-sparse averaging in our case) serves this algorithmic purpose.
>
> We agree with the reviewer that beyond the equilibrium computation purpose, CMWU cannot be viewed as a rational behavioral assumption for selfish agents that seek to minimize their individual cost. However the latter critique holds to a greater or lesser extend for the whole line of research studying uncoupled online learning dynamics (even if their update rule guarantees $\sqrt{T}$ adversarial regret). More precisely, there is no specific reason for a selfish agent to adopt OMWU over any other no-regret algorithm (e.g. Regret Matching), something that would result in the standard $\sqrt{T}$-bound on the regret for all agents.
>
> Even more to this point, if an agent is guaranteed that all their opponents apply OMWU, then she can leverage this knowledge in even more elaborate ways, e.g., to drive the dynamics in a particular part of the state space by incurring short term costs that lead to long term advantages. One such concrete example is shown in [1] the case of auctions with a no-regret buyer, where the auctioneer can provably manipulate the behavior of any buyer learning according to a wide class of no-regret algs such as OMWU so as to guarantee to themselves optimal revenue at the expense of higher prices for the buyer by taking advantage of the commitment of the buyer to a no-regret learning strategy.
>
>
> That being said, we do not claim that ensuring the adversarial no-regret property in the context of uncoupled online learning dynamics (that achieve accelerated convergence rates) is meaningless. However we believe that the learning in games community should discuss and differentiate in greater detail uncoupled online learning dynamics that comes as a natural game-play and uncoupled online learning dynamics that come as means of decentralized efficient computation. We plan to incorporate the above interesting discussion in the camera ready version.
>
> [1] Braverman, Mark, et al. "Selling to a no-regret buyer." Proceedings of the 2018 ACM Conference on Economics and Computation. 2018
>
> > Other works in this space (e.g., the recent paper by Daskalakis et al. on near-optimal regret in games) stressed how the accelerated regret bounds can fall back to the $\sqrt{T}$ usual, non accelerated,  regret dynamics when facing adversarial opponents. Is the same possible for CMWU?
>
> The adversarial no-regret property of OMWU is indeed an additional property that we do not establish for the CMWU dynamics. However CMWU dynamics provides faster convergence rates with a significantly simpler and more intuitive proof than OMWU dynamics. The question of whether the update rule of CMWU can be converted so as to guarantee $\sqrt{T}$ adversarial regret is outside the scope of the current paper, but is nevertheless an interesting question for future research.
>
> > Can CMWU be run as an anytime algorithm, that is, without knowing the value of  a priori? If so, I find that that would be an important addition to the paper.
>
> Yes, it is quite easy to establish the same results by taking $ k = O( \lceil \log (t) \rceil)$ in the update rule of CMWU (Algorithm $2$). We plan to elaborate on this point in the camera ready version.

---

> > ### Comment · Reviewer_UCa9 · 2022-08-09
> > **Thank you!**
> >
> > I thank the authors for the interesting discussion. I think the authors bring up interesting points with regards to the second question, and I tend to agree with them. About the first point, I still encourage the authors to be careful to minimize the possibility of confusion. I maintain my positive impression of the paper.

---

> > > ### Author Response · Authors · 2022-08-09
> > > **Thank you!**
> > >
> > > Thank you for the engaging discussion and the continued support. We will be happy to reflect the main points of this interesting conversation in future versions of our paper. Thank you again!

---

### Official Review · Reviewer_2XD2 · 2022-07-11

**Rating:** 5
**Confidence:** 3
**Soundness:** 2 fair
**Presentation:** 2 fair
**Contribution:** 2 fair

**Summary:**

This paper considers using no-regret algorithms for solving general games. Their proposed algorithm for the players, named CMWU, achieves an O(nVlogmlogT) regret, which improves the previous best results by an O(log^3 T) factor. The main idea is to use future decision to update the current one in MWU.

**Questions:**

The main questions are listed above.

Putting the game framework aside, I am a bit curious about this problem: can CMWU be applied to a learning with expert problem? If so, what is the regret? And what’s the relationship to OMWU?

**Ethics Review Area:**

["I don’t know"]

**Limitations:**

I have some questions about the computational complexity, which are listed above.

**Strengths And Weaknesses:**

Strengths:

1. The proposed algorithm improves the previous state of the art algorithm by an O(log^3 T) factor.

2. The main idea of the algorithm, i.e., using the future to update the current decision in MWU, to my knowledge is novel. I think the main obstacle here is the computational issue, and it seems that the authors have successfully overcome it under certain assumptions. However, I have some concerns about this point, which is listed in the next part.



Weakness:

1. My main concern is about the computational complexity of the proposed algorithm (I realized that the authors say in Line 75 that the update only “requiring a single step of MWU”, so I may have some misunderstanding, and I hope the authors could help me clear the concerns):

1) in general, as the authors pointed out, the CMWU update (Line 173) is very difficult to compute.

2)  Under certain assumptions, the authors show that “the update can be computed linear fast via a contraction map”. However, I have two questions about this conclusion:
a) for the linear convergence rate, do the authors mean one can get a $\epsilon$-approximate solution with O(log (1/epsilon)) number of iterations?
B) If so, will the error  accumulate at each round? And will the real number of iterations become something like TlogT instead of T?

3) I also have a question about the assumption related to the maximum eta. I was wondering if the authors could be more specific about how to set eta to get the regret bound in Table 1?

2. Apart from the improvement on regret, another contribution is the more general deduction rule. Can the authors be more clear about the importance of this contribution? Why is it a significant result?

3. I find the paper is not easy to read, especially to people who are not familiar with the specific area. For example, in the introduction, the authors leave many terminologies unexpaterned, such as mixed strightey and sparse sub-trajectory.

---

> ### Author Response · Authors · 2022-08-01
> **Response to Reviewer 2XD2**
>
> Thank you for your review and feedback, please find our responses to your questions below.
>
> > My main concern is about the computational complexity of the proposed algorithm...
>
> The update rule of CMWU (Equation (CMWU)) is very hard to compute for general $\gamma$. However, we show that if $\gamma \leq \frac{1}{(n-1)V}$, the update rule can be very efficiently computed through the iterations of a contraction map. We refer the reviewer to the update rule of CMWU dynamics (Algorithm $2$) run by each agent that requires only $O(1)$ complexity (at each round $t$). In Theorem 4.1 we show that once one averages the $O(\log T)$-spaced subsequence, the produced average vector converges to CCE.
>
> > a) for the linear convergence rate, do the authors mean one can get a $\epsilon$-approximate solution with O(log (1/$\epsilon$)) number of iterations? b) If so, will the error accumulate at each round? And will the real number of iterations become something like TlogT instead of T?
>
> The reviewer is right that because the error aggregates we take $k:= O(\log T)$. That is exactly the reason why over a $T$ sequence of play, we average the $\log T$-spaced sub-sequence that has length $O(T / \log T)$. This is the reason we get $O(\log T / T)$ rate of convergence, which is exactly what the reviewer writes above (with the $T$ and $T \log T$ equivalence).
>
> > I also have a question about the assumption related to the maximum eta. I was wondering if the authors could be more specific about how to set eta to get the regret bound in Table 1?}
>
> Theorem $4.1$ states that we should set $\eta:= 1/2Vn$. We will add the specific step-size selection to Table $1$ to improve clarity.
>
> > Apart from the improvement on regret, another contribution is the more general deduction rule. Can the authors be more clear about the importance of this contribution? Why is it a significant result?
>
> To the best of our knowledge, prior to our work, the only deduction rules considered formally from a theoretical perspective were either the time-average strategy vector or the last-iterate vector. We find it both interesting and counter-intuitive that an online learning dynamic can achieve better rates by aggregating a subsequence of play. At a conceptual level the latter suggests that specific uncoupled online learning dynamics with deduction rules beyond time-average or last-iterate may achieve better convergence rates.
>
> > I find the paper is not easy to read, especially to people who are not familiar with the specific area.
>
> We will take into account the reviewer's comments and expand on the respective notions in the paper.
>
> > Putting the game framework aside, I am a bit curious about this problem: can CMWU be applied to a learning with expert problem? If so, what is the regret? And what’s the relationship to OMWU?
>
> The update rule of CMWU (Algorithm 2) is an online learning algorithm that can be applied to the expert problem. We believe that the worst-case regret of CMWU is $\Omega(T)$ making CMWU an unappealing algorithm for the adversarial expert problem. OWMU can guarantee $O(\sqrt{T})$ adversarial regret however it admits worse convergence properties in the context of uncoupled online learning dynamics.

---

### Official Review · Reviewer_k1YV · 2022-07-12

**Rating:** 6
**Confidence:** 4
**Soundness:** 3 good
**Presentation:** 3 good
**Contribution:** 3 good

**Summary:**

This paper proposes Clairvoyant Multiplicative Weight Update (CMWU), a simple alternative of Optimistic MWU that can compute CCE of a general-sum game in log(t)/t rates in a uncoupled manner. The idea of the algorithm is to compute the fixed point of the exponential weight update, similar to the idea of proximal-point method. It can be shown that if all players follow this dynamics, the time average of their iterates will be an approximate CCE, with the accuracy of approximation dependent on the step size.

Due to the intractability of computing the fixed point, the authors proposes to approximate it when the step size is smaller than a constant, exploiting the fact that when the step size is sufficiently small, the exponential weight update is a contraction mapping. When every player synchronizes to find the fixed point, they can find the fixed point up to accuracy of 1/t in t steps. The procedure can be decoupled across all players and is very efficient.

Overall, the algorithm can find CCE in a rate of $n\log(m)\log(t)/t$, where $n$ is the number of players and $m$ is the number of pure strategies (for a single player).

**Questions:**

- Please answer the questions asked in the "weakness" part above.
- Corollary 3.3: the $\eta$ should be in the denominator?
- Is there a computational complexity lower bound (might be restricted to some class of algorithms) for the considered setting?


**Limitations:**

There is little/no limitations or potential negative societal impact.

**Strengths And Weaknesses:**

Strength:
- The proposed algorithm has simple implementation and simple analysis, and achieves the state-of-the-art rate of computing CCE in a uncoupled manner.

Weakness:
- The proposed algorithm requires stronger coordination among players compared to OMWU, which might sacrifice some merits of OMWU. For example, the uncoupled version of CMWU requires a synchronization among players in updating z, what if the synchronization fails or the periods used by players differ?
- Similarly, OMWU has the benefit of being a no-regret algorithm against any adversary (if the learning rate is adaptive). Can CMWU also be a no-regret algorithm?

---

> ### Author Response · Authors · 2022-08-01
> **Response to Reviewer k1YV**
>
> Thank you for your review and feedback, please find our responses to your questions below.
>
> > The proposed algorithm requires stronger coordination among players compared to OMWU, which might sacrifice some merits of OMWU...
>
> Exactly as in the case of OMWU dynamics, it is assumed that all players follow the prescribed protocol precisely. For example, if OMWU is implemented with players not updating simultaneously then the convergence properties do not hold. Although the update rule of CMWU is slightly more elaborate than OMWU it does not require any additional synchronization.
>
> > Similarly, OMWU has the benefit of being a no-regret algorithm against any adversary (if the learning rate is adaptive). Can CMWU also be a no-regret algorithm?
>
> The adversarial no-regret property of OMWU is indeed an additional property that we do not establish for the CMWU dynamics.
>
> However CMWU dynamics provides faster convergence rates with a significantly simpler and more intuitive proof than OMWU dynamics. The question of whether the update rule of CMWU can be converted so as to guarantee $\sqrt{T}$ adversarial regret is outside the scope of the current paper, but is nevertheless an interesting question for future research.
>
>
> > Corollary 3.3: the $\eta$ should be in the denominator?
>
> Thank you for catching this typo!
>
>
> > Is there a computational complexity lower bound (might be restricted to some class of algorithms) for the considered setting?
>
> There exists the information-theoretical lower-bound of Daskalakis et al. [11] establishing that any uncoupled online learning dynamics requires at least $T$ payoff-queries so as to converge to a $\Theta(1/T)$-approximate CCE. Notice that CMWU requires $T \log T$ payoff-queries so as to converge to a $\Theta(1/T)$-approximate CCE.

---

> > ### Comment · Reviewer_k1YV · 2022-08-09
> > **about synchronization**
> >
> > Thanks for the detailed response. Here's a little more explanation about the "synchronization" I mentioned in the review (which I did not explain clear enough).
> >
> > By "synchronization", I mean that the current algorithm requires that all players update z only when "t mod k = 0". An "asynchronous" version would be, for example, one player updates it when "t mod k = 0" while another updates it when "t mod k = 1"; or, for example, one updates it when "t mod k = 0" while another updates it when "t mod m = 0" for some m != k. This kind of synchronization is, to my understanding, not required in OMWU.

---

> > > ### Author Response · Authors · 2022-08-09
> > > **Response**
> > >
> > > We thank the reviewer for their comment and their work!
> > >
> > > At an intuitive level CMWU seems to require further sychronization than OMWU. However at a formal level, the implicit "sychronization requirement" is the same in both cases, i.e., all agents follow the prescribed update rule. Under all MWU, OMWU, CMWU rules every agent has a locally computable rule that they apply on every single period to update their behavior and they do so every period. No other synchronization is necessary.
> > >
> > > We should note that the standard analysis of all these algorithms requires that all agents move concurrently/synchronously. If agents are allowed to move asychronously, as the Reviewer suggests, then all standard definitions/analyses no longer apply. To point this out, let's think about regret. Should the agents be penalized for losses occuring when they are not allowed to move? If so how? These are not trivial issues and one could define several different approaches to e.g. standard definitions of regret. See e.g. [1] for an alternative definition of regret applicable to the special case of alternating play in two agents games. Clearly, if one wanted to explore other settings with more agents and different updating schedules they would need new, setting-appropriate definitions and analyses. We hope that this clarifies the issue but we are happy to discuss more, if needed.
> > >
> > > [1] Bailey, James P., Gauthier Gidel, and Georgios Piliouras. "Finite regret and cycles with fixed step-size via alternating gradient descent-ascent." Conference on Learning Theory. PMLR, 2020.

---

### Author Response · Authors · 2022-08-07
**Thank you for your work. Any more feedback for us?**

Dear Reviewers,

Thank you very much again for your time for reviewing our paper and for your helpful comments and suggestions. We were wondering if there is anything you would like to discuss. If you have any remaining questions/suggestions, we would be more than happy to discuss them further with you.

Thank you again!

---

### Author Response · Authors · 2022-08-09
**Updates on Experimental Results**

Dear Reviewers, thank you for your time and effort reviewing our paper.

We would like to update you that in accordance with the wishes of Reviewer CoXh, we have performed experiments comparing CMWU and the state-of-the-art implementation of OMWU and observed experimentally that as our theory suggests, CMWU allows for faster CCE computation than OMWU, i.e. CMWU requires significantly less oracle calls than OMWU to reach the same $\epsilon$-CCE. We have added these experiments as a supplementary section in our paper, and would be happy to add more experiments as the reviewers see fit. Thank you!

---

### Meta-Review · Area_Chair_3umg · 2022-08-27

**Recommendation:** Accept
**Confidence:** Less certain

**Metareview:**

This paper proposes a new uncoupled method for computing CCE of a general-sum normal-form game with a SOTA rate.
As the reviewers point out, the biggest issue of this work is that the proposed algorithm is not no-regret considering all iterates. We strongly encourage the authors to make this clearer and avoid making misleading/confusing statements. Reviewer k1YV's concern on the "synchronization" requirement is also a very valid point, and the authors' response did not really address this concern. We encourage the authors to also make this clearer in the revision.

**Award:**

No

---

### Decision · Program_Chairs · 2022-09-14

Accept